# Phytochemicals Profiling, Antimicrobial Activity and Mechanism of Action of Essential Oil Extracted from Ginger (*Zingiber officinale* Roscoe cv. Bentong) against *Burkholderia glumae* Causative Agent of Bacterial Panicle Blight Disease of Rice

**DOI:** 10.3390/plants11111466

**Published:** 2022-05-30

**Authors:** Mahesh Tiran Gunasena, Amara Rafi, Syazwan Afif Mohd Zobir, Mohd Zobir Hussein, Asgar Ali, Abdulaziz Bashir Kutawa, Mohd Aswad Abdul Wahab, Mohd Roslan Sulaiman, Fariz Adzmi, Khairulmazmi Ahmad

**Affiliations:** 1Department of Plant Protection, Faculty of Agriculture, Universiti Putra Malaysia, Serdang 43400, Malaysia; gs57269@student.upm.edu.my (M.T.G.); gs60630@student.upm.edu.my (A.R.); abashir@fudutsinma.edu.ng (A.B.K.); mohdaswad@upm.edu.my (M.A.A.W.); 2Grain Legume and Oil Crop Research and Development Centre, Angunakolapelessa 82220, Sri Lanka; 3Institute of Nanoscience and Nanotechnology, Universiti Putra Malaysia, Serdang 43400, Malaysia; mzobir@upm.edu.my; 4Centre of Excellence for Postharvest Biotechnology (CEPB), School of Biosciences, University of Nottingham Malaysia, Semenyih 43500, Malaysia; asgar.ali@nottingham.edu.my; 5Department of Plant Science and Biotechnology, Faculty of Life Science, Federal University Dutsin-Ma, Dutsin-Ma 821101, Nigeria; 6Department of Biomedical Science, Faculty of Medicine and Health Sciences, Universiti Putra Malaysia, Serdang 43400, Malaysia; mrs@upm.edu.my; 7Institute of Plantation Studies (IKP), Universiti Putra Malaysia, Serdang 43400, Malaysia; farizadzmi@upm.edu.my; 8Institute of Tropical Agriculture and Food Security (ITAFoS), Universiti Putra Malaysia, Serdang 43400, Malaysia

**Keywords:** antimicrobial activity, bioactive compounds, *Burkholderia glumae*, gas chromatography, ginger essential oil

## Abstract

Essential oils protect plants, and due to their natural origin, there is much interest in using them as antimicrobial agents. The purpose of this study was to determine the phytochemical constituents of ginger essential oil (GEO), antimicrobial activity, and mode of action against *Burkholderia glumae* (*Bg*). In addition, the volatile active compounds (AIs) were studied using GC-MS, FTIR, and Raman spectroscopy. A total of 45 phytochemical components were detected and the most prevalent bioactive compounds were Geranial, 1,8-Cineole, Neral, Camphene, α-Zingiberene, and α-Farnesene. Furthermore, it was found that the most dominant terpenes in GEO were monoterpenes. The diameter zone of inhibition values varied from 7.1 to 15 mm depending on the concentration tested. In addition, the MIC and MBC values were 112.5 µL/mL. Faster killing time and lower membrane potential were observed in 1xMIC treatment compared to 0.5xMIC treatment, whereas the control had the maximum values. From observations of various images, it was concluded that the mode of action of GEO affected the cytoplasmic membrane, causing it to lose its integrity and increase its permeability. Therefore, the antibacterial study and mechanism of action revealed that GEO is very effective in suppressing the growth of *B. glumae*.

## 1. Introduction

*Burkholderia glumae* is a seed-borne rice pathogen and the most serious emerging pathogen in paddy-growing countries. The bacterial panicle blight (BPB) of rice caused by *Bg* can induce up to 75% yield loss in a severely infested field [1,2]. The *Bg* is Gram-negative, non-spore-forming, nonfluorescent, and rod-shaped with a polar flagellum bacterium [2,3]. The disease causes symptoms such as blight in rice panicles, discoloration of growing grains, seed rot, and spikelet sterility [3,4]. *B. glumae’s* incidence in plumules results in the production of toxic materials such as toxoflavin. The toxoflavin [1,6-dimethylpyrimido [5,4-e]-1, 2,4-triazine-5,7(1H,6H)-dione] is a yellow pigment required for pathogenicity and causes rice seedling rot [2]. Weather conditions, including excessive daily minimum temperatures and regular rainfall throughout rice panicle appearance and flowering stages, which increase the relative humidity conditions, have a considerable impact on BPB disease outbreaks [5]. *B. glumae* is a good example of minor plant pathogens progressing to a serious disease as environmental conditions change. Similarly to how it affects rice, the *Bg* also affects pepper, eggplant, sesame, and tomato plants [2]. Plant extracts, resistant varieties, seed treatments, and heating technologies that have been proven to inhibit *Bg* infections are among the most common methods used to overcome the disease [6].

Recently, significant control efforts have been made to limit pathogen dissemination using numerous control methods, such as the use of substitute antimicrobial compounds. Essential oils (EOs) as natural antimicrobial agents have a role in plant defense mechanisms, and they have a high probability of being used for controlling plant pathogens [7]. Aromatic volatile compounds derived from various plant parts were extracted by numerous techniques and are known as EOs [8,9]. For the commercialization process, a conventional hydrodistillation method that utilizes water that leaves nontoxic byproducts is the most effective method for EO extraction [10]. The EOs and their constituents have a long history of use as antimicrobial agents. This biological activity is frequently attributed to bioactive components developed by EOs during the secondary metabolism of the plant. The most EOs are classified into two types based on the number of isoprene units: monoterpenes and sesquiterpenes [11]. Monoterpenes are terpenes made up of two isoprene units that are either linear (acyclic) or comprise of rings (monocyclic and bicyclic). Sesquiterpenes are types of terpenes composed of three isoprene units [12].

Ginger (*Zingiber officinale* Roscoe) is one of the most widely known perennial herbs. It is derived from the underground stems or rhizomes of the plant [13]. In Malaysia, ginger is cultivated commercially in Pahang, Johor, Selangor, Sabah, and Sarawak. *Zingiber officinale* cv. ‘Tambunan’ is a popular cultivar of ginger in Sabah, and is mainly cultivated in the Tambunan and Keningau areas. *Z. officinale* cv. Bentong is the most expensive and popular ginger cultivar in Malaysia, and is mainly cultivated in Bentong, Pahang. The Zingiberaceae family is amongst the most commonly grown plant families in the tropics, specifically in Southeast Asia. This valuable natural resource produces a variety of useful items, such as food, spices, medications, colors, perfumes, and cosmetics. Ginger is widely used as a spice and flavoring agent around the world and is also used medicinally [13,14]. Additionally, Zingiberaceae species have been identified to have broad biological actions including antimicrobial, antioxidant, and anti-inflammatory properties. Ginger essential oil is a volatile oil obtained from the root of ginger and is known for its distinct aroma and biological activity [10]. Ginger has been extensively studied and it contains 1-2% EO, which gives the spice its distinct flavor [15]. As a result of the increasing attention to the usage of EOs in the control microbes, GEO has been studied for its role against different pathogens. Conventionally, chemical components of EOs can be detected and measured using chromatographic methods, which generate high-quality and consistent results. Raman spectroscopy procedures can offer important analytical information in significantly less time than conventional methods. There are numerous studies on the chemical content of fresh GEO and other natural flavoring compounds [15]. Terpenes, phenolics, and N- and S-containing derivatives are the three most common types of secondary metabolites produced by plants [16]. Ginger contains many active ingredients, like phenolic and terpene compounds. In vitro antimicrobial activity of EOs has been widely studied. Furthermore, it is a rich source of several active substances, such as bioactive phenols (gingerols, shogaols, and zingerones) [9]. It also contains several terpene components, including β-bisabolene, α-curcumin, zingiberene, α-farnesene, and β-sesquiphellandrene, which are thought to be the major components of GEOs [17]. More importantly, recent work emphasizes the contribution of the phenolic compounds (Eugenol, Shogaols, Zingerone, Gingerdiols, Gingerols, and so on) and their synergistic interactions with other substances such as β-Sesquiphellandrene, α-Zingiberene, α-Farnesene, α- and β-Bisabolene found in GEO, extracts, and Oleoresins, which played a crucial role as the antimicrobial agent [18]. Gram-negative bacteria are commonly more opposed to EO than Gram-positive ones. Gram-negative bacteria have an outer membrane made of hydrophilic lipopolysaccharides (LPS), which operate as a barrier against hydrophobic chemicals like those contained in EOs [19]. Moreover, GEO has also demonstrated the ability to suppress bacterial growth due to the reduction of bacterial biofilm formation [17]. GEO has been widely studied, with particular attention to its antifungal, antibacterial, and antioxidant properties, in addition to its potential use in preserving food [20]. The advantages of using GEO were highlighted in numerous studies, which demonstrated its favorable impacts against disease symptoms, acting as an anti-inflammatory, antitumor, anodyne, neuronal cell-protective agent [21]. Therefore, the primary objective of this study was to determine the phytochemical profiling, antibacterial activity, and mechanism of action of GEO against *B. glumae.*

## 2. Results

### 2.1. Analysis of Chemical Constituents of Ginger Essential Oils by Gas Chromatography-Mass Spectrometry 

The results of the gas chromatography–mass spectrometry (GC–MS) study revealed various bioactive compounds identified and quantified in GEO. The chemical components of GEO through hydrodistillation was listed in Table 1 with their respective percentages. EO constituents were identified by comparing retention times and retention index with library data from FFNSC1.3.lib, NIST11.lib, and WILEY229. The retention index was calculated using the Kovats equation. In addition, the comparison of retention values obtained by Sharma et al. (2016) [15] and Babushok et al. (2009) [22] is also provided in Table 1.

The extraction yields were determined by dividing the amount of extracted oil by the amount of fresh material fed into the extractor. The extraction yields were estimated based on the results of several cycles. The results were presented as a percentage of extraction. The yield of GEO was between 1.2–1.5% (*v*/*w*). The EO was extracted from the ginger rhizome in which the color was changed from pale yellow to light amber. Gas chromatography analysis identified 45 peaks of active compounds available in GEO as displayed in Figure 1 It was also observed that the major compounds were Geranial (17.88%), 1,8-Cineole (14.96%), Neral (13.99%), Camphene (7.53%), and α-Farnesene (3.51%) and their combination was more than 50%. The GEO contained a high percentage of monoterpenes (74.6 %) and sesquiterpenes (20.73%). Geranial was the most prevalent monoterpene hydrocarbon compound in the total extracted essential oil. Among the 20 monoterpenes, the major ones were Geranial, 1,8-Cineole, Neral, Camphene, Citronellol, α-Pinene, and β-Linalool. The remaining monoterpenes were found in less than 2% of the samples. The predominant sesquiterpenes were α-Zingiberene (5.19%) α-Farnesene (3.51%), β-Sesquiphellandrene (2.40%), Curcumene (1.58%), and β-Bisabolene (1.16%). Less than 1% of the other sesquiterpenes were detected in the sample. Our findings are in line with the findings of Abdullahi et al., (2020) [23]. 

### 2.2. Vibrational Spectroscopy Analyses by Raman and FTIR

Both vibrational spectroscopy analyses of Raman and Fourier transform infrared (FTIR) spectra are displayed in Figure 2. The appearance of several peaks ranging from 400 cm^−1^ to 3000 cm^−1^ was attributed to the existence of multiple chemical bonds in the GEO (Figure 2a). The most important components in ginger essential oil are zingiberene, geranial, neral, and camphene [24]. At the lower frequency of the Raman spectrum (700–900 cm^−1^), the COC and CCO bands were detected. The CH-stretching band was at 1438 cm^−1^. Other appearance peaks for benzene rings were recorded at the middle range frequency: 1590 and 1602 cm^−1^ of the Raman spectrum. The C=C band was at 1642 and 1684 cm^−1^, which attributed to the CH_3_ group bending mode. In the chemical structure of geranial and neral, the double bond is conjugated with the aldehyde group. CH-bending mode was also detected at 2800–2841 cm^−1^. The Fourier Transform Infra-Red spectrum was recorded from 550–4000 cm^−1^, as displayed in Figure 2b. Plenty of functional groups were detected, such as OH bonds, COC, C-C stretch, C=O stretch, and OH stretch [25]. Detailed information about the peaks, wavelength, and functional groups was tabulated in Table 2. 

### 2.3. Antibacterial Activity, Minimum Inhibitory Concentration (MIC), and Minimum Bactericidal Concentration (MBC) of Ginger Essential Oil 

Ginger essential oil antibacterial activity was assessed qualitatively by measuring the diameter of the zone of inhibition (ZOI). Evaluation of the antibacterial activity of GEO was recorded in Table 3. The negative control was dimethyl sulfoxide (DMSO) buffer, and the positive control was streptomycin. The results revealed that all concentrations were potentially effective in suppressing the growth of *Bg* with variable potency. The widths of the inhibition zones increased as the concentrations of GEO increased. The bacterial sensitivity to GEO was ranked using zone-of-inhibition values expressed in millimeters (mm) [26]. The antibacterial activity of 100, 200, 300, 400, and 500 µL/mL of GEO against *Bg* indicated inhibition zone diameters of 7.1, 8.0, 10.6, 14.3, and 15.1 mm, respectively. It was found that the most effective retarding potent antibacterial action against *Bg* was at 500 µL/mL. These treatments incited sensitive reactions and a high percent of inhibition of growth. Compared to a standard positive control used in the experiment, the ZOI of streptomycin was 24.1 mm. The negative control DMSO demonstrated no inhibition of the growth of *Bg.*

The antibacterial effect of GEO was quantified using MIC and MBC values against *Bg*. The color appearance of the well contents was determined by the inhibition of *Bg* at various concentrations. The wells containing nonviable bacterial cells demonstrated no color change, whereas the wells containing viable cells presented red color formation. The MIC and MBC were determined to see if GEO could inhibit or kill the pathogenic bacteria *Bg*. The results for the MIC determination in this study revealed that GEO has potent antibacterial activity against *Bg*. It was critical to determine the efficient concentration in vitro. GEO had an MIC of 12.5 L/mL against *B. glumae* and the MBC value for *B. glumae* was 12.5 L/mL. The MIC and MBC for the GEO were 12.5 µL/mL, it served as the concentration whereby approximately all the *Bg* cells were killed. The MBC value for *Bg* was the same as the MIC value.

### 2.4. Time Killing Analysis 

A bacterial growth-curve assay was carried out to determine the efficacy of GEO in killing or inhibiting the growth of *Bg*. The assessment of GEO inhibition or deactivation kinetics on *Bg* gives supportive evidence that supplements the data acquired from the MIC and MBC investigations, permitting for a more accurate estimation of the component efficiency. The analysis of *Bg* under various concentrations of GEO was illustrated in Figure 3. The time courses of *Bg* growth in the presence of different concentrations of GEO were plotted to confirm the antibacterial action of GEO against *Bg*. 

The optical density values of the control treatment increased rapidly during 2–18 h of incubation periods. When compared to the 0.5xMIC of GEO, the optical density of *Bg* slightly increased from 2–8 h, which was mostly attributable to the bacteria’s growth cycle inhibition. Interestingly, pathogen growth was slowed down with an extended lag time when the concentration of GEO approached 1 × MIC. The optical density dropped from 0–4 h of incubation periods and the optical density value remained constant onward. Compared to the control, susceptible *Bg* treated with GEO at 1 × MIC level demonstrated a significant decrease in the number of the viable cells during the first 6-8 h of the assessment periods and needed up to 14 h of incubation to promote the entire elimination of *Bg* cells in this assay.

### 2.5. Morphological Changes of B. glumae by Scanning Electron Microscope (SEM) 

The scanning electron microscope is commonly applied to monitor the external morphological deviations of the cells. The physical and morphological variations of *Bg* at different treatments were examined by SEM images, as demonstrated in Figure 4. SEM images revealed that the treated *Bg* surfaces underwent obvious morphological changes compared to the DMSO-treated ones (negative control). Figure 4a illustrates the morphology and microstructure of *Bg* exposed to DMSO, with distinguishing features such as normal rod-shaped, undamaged surface, striated cell walls, and no visible cellular debris. However, after 24 h of exposure to GEO at MIC (Figure 4b), morphological deviations of *Bg* were observed as distorted, irregular, and shriveled to different degrees. The bacterial colonies exposed to GEO were severely damaged, with contraction illustrated by the yellow arrows in Figure 4b. The cells incubated with streptomycin presented collapsed and ruptured cells in Figure 4c. Streptomycin treatment resulted in more severe morphological destruction than GEO treatment. This was most likely owing to the severely damaged bacterial peptidoglycan layer and cell membrane, which resulted in cell death and disconnection from the filter holder.

### 2.6. Morphological Changes of B. glumae by Transmission Electron Microscope (TEM)

The effects of GEO as an antimicrobial against *Bg* cells were further investigated using TEM, which determines the cell membrane integrity and internal morphology of the cells, and the results are displayed in Figure 5. The images of the *Bg* cells treated with GEO presented considerably differently from those of the untreated cells. TEM observations revealed that the cells treated with DMSO remained unchanged, with distinct, uniformly shaped, and intact cell walls, evenly dispersed cytochylema, and electron-dense substance within the cell (Figure 5a). However, there were lesser-intact *Bg* cells in the GEO treated samples, and it was difficult to identify the cells that had been exposed to the GEO (MIC). After being exposed to GEO for 24 h, the *Bg* cell was distorted and crumbled, demonstrating that the intracellular contents were leaked out of the cells (Figure 5b). Furthermore, the cell wall and cytoplasmic membrane had become irregular and thick; much lysis was observed. This has the potential to cause nutrients and genetic substances to be discharged from the cell.

### 2.7. Biofilm Breakdown Observed by Confocal Laser Scanning Microscope (CLSM)

Confocal laser scanning microscopy studies were used to investigate the bactericidal mode of action of the active compounds found in EOs. The fluorescence intensity of the stained (red or green) pathogen cells at a specific wavelength indicated the findings of the LIVE/DEAD analyses. The results demonstrated that after staining with the LIVE/DEAD stain, there were no detectible dead cells in the untreated control, having viable *Bg* cells. In the untreated control, the biofilm aggregation size appeared in a green color, indicating the presence of live and viable pathogen cells (Figure 6(a1)). Conversely, the images for the treated *Bg* cells with the MIC of GEO have exhibited membrane damage (Figure 6(b1)), with red staining confirming the existence of many dead cells. These findings revealed that GEO at MIC inhibited and broke down the produced biofilm, limiting its development and growth. The effect of GEO on *Bg* biofilm was further analyzed using 3D confocal microscopy software (Figure 6(a2,a3,b2,b3)). Comparatively, the untreated *Bg* biofilm with GEO images was displayed in Figure 6(a2,a3). Volumetric images revealed a significant reduction in *Bg* biofilm and a higher proportion of damaged/compromised cells (Figure 6(b2)). As depicted in Figure 6(b3), GEO manages to kill *Bg,* starting from the top and working down to the lower layers of the biofilm cells. Furthermore, fewer viable cells can be observed by the green color of the treated biofilm cells. 

## 3. Discussion

The composition, structure, and functional groups of essential oils play an important role in determining their antimicrobial activity. GEO is a complex combination of volatile compounds formed as secondary metabolites in ginger that have been extensively utilized as possible substitutes for chemically produced antimicrobials and antioxidants [27]. Many analyses have stated the chemical composition of ginger essential oil [13,23,28,29]. However, the specific content of the extraction and the number of compounds in plant essential oils are varied and depend on the environmental factors; the growing season of plant, cultivation type, vegetative stage [30], harvesting period, plant part utilized, and method of extraction [21]. The 45 chemical compounds of ginger essential oil were identified in this study. Guimaraes et al. (2019) [31] reported that essential oils are composed of a complex mixture of compounds ranging from 20 to 60 at different concentrations. Based on the GC–MS analysis, the highest percentages of active compounds in GEO were Geranial, 1,8-Cineole, and Neral compounds (Table 1) which agrees with the findings of Mesomo et al., (2013) [21]. The work highlighted the GC–MS analysis of GEO AIs and concluded that the main volatile compounds existing in fresh ginger rhizome were Geranial, ar-Curcumene, and β-Bisabolene. In addition, according to Abdullahi et al. (2020) [23], sesquiterpenes and monoterpenes were found in EOs, which contributed to significant antimicrobial activity against pathogenic microorganisms. In concurrence with previous work conducted by Wang et al. (2020) [9], the essential oil contained a high concentration of monoterpenes, and was then followed by sesquiterpenes.

Those molecular structures are Geranial, 1,8 Cineole, Neral, Camphene, α-Zingiberene, and α-Farnesene, in addition to the combination of those AIs have 60% of GEO compositions. Next, the physicochemical characterizations were done using vibrational spectroscopy analyses, such as Raman and FTIR spectra, as demonstrated in Figure 2. In general, all AIs Have C=C, C-C, and C-H bonding, which was recorded on both spectra. However, the existence of benzene rings is exclusive only for 1,8 Cineole and α-Zingiberene which were recorded on the Raman spectrum at 1590 and 1602 cm^−1^. 

The analysis of ZOI, MIC, MBC, and the time-killing analysis of *Bg* were studied to assess GEO antibacterial action. The EOs and their main constituents have demonstrated a high level of antimicrobial activity against pathogens [32]. Similarly, because of the high content of lipophilic compounds, EOs attached and interrupted the integrity of microbial cell wall surfaces and membrane structures, leading to cell lysis [33]. The antibacterial activity against *Bg* demonstrated varying degrees of growth inhibition, which were highly dependent on the concentration of GEO. A wider ring (higher ZOI) of no bacterial growth occurred because of a stronger inhibitory effect of antimicrobial EOs, whereas a lesser or no inhibition action results in few or no changes in the bacterial cell membrane. The EO concentration (100–500 µL/mL) affected the bacterial growth of *Bg*. The MIC study strongly indicates that GEO has potent antibacterial activities against *Bg*. In this circumstance, the MBC value was the same as the MIC value for *Bg* which was consistent with the time-killing study. The information clearly indicated that the GEOs at 1 × MIC could kill large amounts of *Bg* at different times compared to GEO at 0.5 MIC. Sharma et al. (2016) [15] found that GEO had an inhibitory action on a broad range of pathogenic bacteria and fungi, which was most likely due to the oil’s major components. Furthermore, the surface of Gram-negative bacteria cells acts as a penetration barrier, preventing macromolecules and hydrophobic compounds from penetrating the bacteria cell membrane [27]. As a result, Gram-negative bacteria are more tolerant to hydrophobic antibiotics than Gram-positive bacteria. However, Wang et al. (2020) [9] discovered that GEO had antibacterial properties against both *Escherichia coli* and *Staphylococcus aureus*. The findings of the MIC study and the growth curve from our investigation both pointed to the same conclusion. GEO and extracts were found to have great antimicrobial action and inhibited some food-spoilage microorganisms [34]. Our findings confirmed that GEO can significantly inhibit *Bg* growth by providing a theoretical basis for effective antibacterial activity. According to Dannenberg et al.’s (2018) [35] time-killing analysis, after 8 h of making contact with rosemary EOs (MBC = 0.1%), the total mortality of the bacteria (*S. aureus*) was observed while the main component (1,8-Cineole) was treated separately and needed 12 h. The differences were more accessible, which was similar to the findings of the SEM and the kill-time study.

Based on the different mechanisms of action, the antimicrobial substances on the membrane could cause depolarization or increased permeability. For example, several antimicrobial peptides form pores, whereas other compounds, such as certain EO constituents, have a fluidifying effect on the membrane [36]. SEM, TEM, and CLSM assays were utilized to discover the antibacterial mode of action of GEO against *Bg*, with the MIC concentration of GEO causing changes in cell viability, morphology, membrane leakage, and cell integrity. The integrity of the cell is critical for an organism to continue to exist because it is the main component for critical biological activities that take place [37]. Despite this, certain mechanisms have been suggested. The cell membrane breaks down and changes in ion channels (Na^+^, K^+^, Ca^2+^, or Cl) in the cell membrane. This could enhance permeability and cause the leakage of important intracellular substances [38] in addition to the inhibition of target enzymes [39]. The morphological deviations from the images clearly demonstrated that the cell membranes of the *Bg* treated with GEO were significantly disrupted, with noticeable abnormal structure and shape. Most importantly, GEO demonstrated the ability to enhance the permeability of cell membranes by causing the leakage of intracellular substances, which was thought to be a mechanism of the *Bg* downstream physiological phenomena. Most of the treated GEO bacteria formed irregular, sunken surfaces and shriveled to varying degrees. This study discovered similar research findings in bacterial morphological changes as other studies [20,40,41]. Next, CLSM analysis revealed that the GEO inhibited the treated cell, resulting in a notable breakdown and reduction in biofilm development. As a result, GEO decreased the quantity of viable *Bg* cells, and the *Bg* growing medium made the conditions unfavorable for biofilm development. The assay demonstrated a reduction of entire structures and cell density when compared to the control, where the living cells seemed to be embedded in the polysaccharide matrix. As previously reported by Kerekes et al., (2013) [42], the nonviable cells were surrounded by a layer of living cells. 

We may make a preliminary conclusion, based on the results of transcriptional profiling tests, that GEO could be useful in treatment strategies, based on the considerable bacterial reduction/inhibition of *Bg* pathogen (in vitro). However, our knowledge of biocontrol for the individual bioactive compounds in the EO has some limitations. As a result, future studies should focus on the mechanisms of action of individual EO active compounds, and a systematic investigation into the mechanisms of synergistic interaction between compounds should be encouraged.

## 4. Materials and Methods

### 4.1. Ginger Essential Oil Extraction

The essential oil from fresh ginger rhizomes was extracted by utilizing a Clevenger-type apparatus and the hydrodistillation method [15]. The fresh rhizome of ginger (*Z. officinalae* cv. Bentong) (Voucher no: KM0006/22) was obtained from a commercial ginger farm in Bentong, Pahang, Malaysia, and the botanical name was confirmed by Dr. Khairil Mahmud of the Institute of Biological Sciences, Universiti Putra Malaysia (UPM). First, the rhizomes were cleaned with water to remove inert materials, then cut into slices and crushed into small particles using a blender. The ground ginger rhizomes were mixed in a 3:5 ratio with distilled water and boiled directly in the 10 L flask for 5 h. The aromatic compounds in the rhizome were let off by the hot steam. Then, the molecules of these volatile compounds were discharged from the plant parts and evaporated into the vapor inside the system, while the temperature of the steam was properly controlled. The sample was heated for 20 min at 100 °C, then decreased to 45 °C for 4 h for an effective extraction process. Finally, the extracted essential oil was placed in a 15 mL Falcon tube and kept in a refrigerator at 4 °C. According to Lucero et al., (2009) [43], the following equation was used to calculate retention indices of GEO.
(1)$$Ii=100\left[n+\frac{\log\left(t_{i}-t_{0}\right)-\log\left(t_{n}-t_{0}\right)}{\log\left(t_{n+1}-t_{0}\right)-\log\left(t_{n}-t_{0}\right)}\right]$$
where the variables used are:

*I**_i_*, the retention index of peak *i*

*n*, the carbon number of *n*- alkane peak heading peak *i*

*t**_i_*, the retention time of compound *i*, minutes

*t*_0_, the air peak, void time in average velocity u = L/*t*_0_, minutes.

### 4.2. Analysis of the Chemical Compound in Ginger Essential Oil

This study was performed to assess the volatile composition and its abundance in GEO. The GC–MS Shimadzu QP-2010 Ultra was used to analyze the volatile compounds according to the method of Wang et al. (2020). [9] The GC–MS system consists of a gas chromatograph interfaced with a mass spectrometer and an SLB-5ms capillary column (30 m × 0.25 mm I.D. × 0.25 µm film thickness). The temperature of the oven was initially fixed at 50 °C (isothermal for 3 min) with a 10 °C/min increase, then increased to 250 °C for 10 min, and finally to 300 °C for 10 min isothermal. At 70 EV, a scan interval of 0.1 s was used, and fragments ranging from 40 to 700 Da were recorded. The total time spent running the GC was 45 min. In this study, helium (99.999 %) was used as the carrier gas, with a flow rate of 0.8 mL/min, an injection volume of 1 µL, a split ratio of 10:1, and an injector temperature of 250 °C. The mass spectrum of ginger constituents was determined by matching retention times, retention index, and mass spectral data from FFNSC1.3.lib, NIST11.lib, and WILEY229.LIB as well as with available literature [9,15,29]. The proportion of the respective compound was indicated as a percentage of the peak area in comparison to the total peak area. The chemical compound, proportions, retention times, retention index, and molecular formulae of chemical compounds were determined.

### 4.3. Antibacterial Activity of Ginger Essential Oil

The antibacterial activity of GEO was evaluated against *Bg,* which is causing bacterial panicle blight disease in rice. In this study, the bacteria stock culture was obtained from the Bacteriology Laboratory at the Department of Plant Protection, Faculty of Agriculture, Universiti Putra Malaysia. The methodology for analyzing GEO antibacterial action was carried out by applying the standard agar disc diffusion technique, also known as the Kirby–Bauer antimicrobial susceptibility test, as described by Tu et al., (2018) [44] with a few modifications. The KB agar was used for the antibacterial activity test. Initially, 20 mL of KB agar was poured aseptically into a sterilized Petri plate and left for 30 min to allow the agar to solidify. Bacteria were subcultured overnight in KB agar at 37 °C, and bacterial growth was harvested using sterile distilled water. Using a Multiskan GO microplate spectrophotometer (Thermo Fisher Scientific, Vantaa, Finland), bacteria culture was diluted with distilled water to achieve an optical density of 0.15 OD 600 nm (1 × 10^8^ CFU/mL) for the *Bg* bacteria. Then, using a sterile L-shaped glass road, 100 µL of the bacterial suspension was evenly spread on agar plates and allowed to dry for 5 min. The GEO were dissolved in DMSO (at different concentrations of 100, 200, 300, 400, and 500 µL/mL). A Whatman No.1 sterile filter paper disc (6 mm diameter) was loaded with 10 µL/disc of different concentrations of GEO. Streptomycin sulfate (0.15 mg/mL) was applied as a positive control, and DMSO was used as a negative control. The plate was left at room temperature for 30 min to allow the oil to diffuse into the agar. After that, the discs were aseptically placed on the inoculated culture plates with sterile forceps, and finally, the plates were incubated at 37 °C for 24 h. The diameter of the ZOI against *Bg* was measured in millimeters to assess the antibacterial activity. The test was carried out in triplicate. According to Khalid et al., (2017) [45], the following equation was used to calculate the inhibition percentage of GEO at each concentration.
(2)$$Percent\;of\;Inhibition=\frac{Zone\;of\;inhibition\;of\;test\;sample\;\left(mm\right)}{Zone\;of\;inhibition\;of\;standard\;positive\;control\;\left(mm\right)}\times 100$$

### 4.4. Assessment of MIC and MBC

The GEO MIC was determined using the method described by Al-Shuneigat et al. (2020) [46] with modifications. The MIC of GEO was determined by applying a twofold dilution assay in 96-well flat-bottom microtiter plates with liquid cultures. The EO were diluted to the subsequent serial dilutions: 20%, 10%, 5%, 2.5%, 1.25%, 0.62%, 0.31%, 0.15%, and 0.07%. Sterile MHB (150 µL) was pipetted into each well of the microtiter plate except for well number 10. The first column was then filled with 150 µL of diluted GEO (400 µL/mL), and serial dilution was carried out until well number 9 was reached. A 150 µL of well number 9 content was thoroughly mixed, and half of the volume was dispensed. In each well, an equal volume of 50 µL *Bg* suspension with an OD 600 nm spectrophotometer value of 0.15 was added. Only MHB and *Bg* suspension was used as a negative control in well number 11. Positive control (well 12) contained MHB, *Bg* suspension, and streptomycin. The plates were incubated for 24 h at 37 °C. A quantitative tetrazolium-based colorimetric method was used to determine the MIC. In each well except well number 10, 50 µL of 1.5% 2,3,5-tetrazolium chloride (TTC) was added. Then, the plate was incubated for 24 h at 37 °C inside the incubator, and the color change from pale yellow to pinkish red was considered positive. The test was performed in triplicate. The MIC was decided by choosing the lowest concentration (highest dilution) of GEO that fully inhibited the growth of the bacteria after 24 h. 

MBC was defined as having the greatest dilution (the lowest concentration) and no bacterial growth on the plates. 100 µL of clear wells were cultured on KB medium plates using the drop plate procedure, and the plates were incubated at 37 °C for 24 h. The MBC value was interpreted as no bacterial growth on the KB medium plate. For each concentration, the tests were conducted in triplicate.

### 4.5. Time Killing Analysis

The logarithm of the relative population size as a function of time was used to create the growth curve, which typically illustrated the bacteria’s vigor. The antibacterial activity was determined by calculating the bacterial growth curve using the method presented by Cai et al., (2018) [47] with minor modifications. *Bg* suspension (1 × 10^7^ CFU/mL, 0.1 mL) was inoculated into a 100 mL sterile MHB medium in a culture flask, and ½ × MIC and 1 × MIC concentrations of GEOs were added to the bacterial suspension, respectively. For the control treatment, the same volume of DMSO was used in the MHB medium. The cultured broths were then incubated for 24 h at 37 °C and shaken at 180 rpm in a rotary shaker. The turbidity of the culture was measured every 2 h by measuring the optical density OD 600 nm for 24 h with a spectrophotometer. To determine an overall average, each set of tests was replicated three times.

### 4.6. Morphological Changes Observed by Scanning Electron Microscope

Scanning electron microscope studies were carried out to observe the morphological deviations of *Bg* treated with GEO, as described by de Oliveira et al., (2011) [48,49] with few adjustments. An SEM study was performed at the microscopy unit, Institute of Bioscience (IBS), UPM. The *Bg* was cultured in KB Petri plates at 37 °C for 24 h and used for SEM. The area of the ZOI around the disc treated with the GEO MIC concentration, Streptomycin, and DMSO (without inhibition) was cut into a small piece 10 × 10 mm in size with a sterile scalpel. All the test samples were subject to primary fixation with 4% glutaraldehyde at 4 °C for 2 h. Then the treated samples were rinsed three times with 0.1 M sodium cacodylate for every 10 min interval. For the post-fixation, the samples were treated with 1% osmium tetroxide at 4 °C for 2 h. The samples were washed three times at an interval of 10 min with 0.1 M sodium cacodylate. Subsequently, the samples were dehydrated with the sequence of acetone in ascending order (35, 50, 75, and 95%) for 10 min and 100% for 15 min 3 times. After completion of the dehydration, the specimens were transferred into a specimen basket and placed into the critical dryer for 90 min. The samples were adhered onto the stub using clean double side tape. Finally, the samples were coated with gold particles and ready for viewing.

### 4.7. Morphological Damage Observed by Transmission Electron Microscope (TEM)

A transmission electron microscope was used to determine the structural deviations of *Bg* bacterial cells, as reported by Sahu et al. (2018). [41] The *Bg* was incubated in MHB and then treaded with GEO MIC concentration, and DMSO, respectively. Following a 24 h incubation period at 37 °C, *Bg* cells were obtained by centrifuging them at 5000 rpm for 5 min and then washing them three times in PBS to eliminate any media or other materials. The bacteria pellet was then prefixed for 6 h at 4 °C with 4 % (*v*/*v*) glutaraldehyde. Following that, samples were rinsed three times in 0.1 M sodium cacodylate buffer for 10 min each. The specimens were post-fixed for 2 h at 4 °C in 1% osmium tetroxide. The samples were rinsed three times in 0.1 M sodium cacodylate buffer for 10 min each time and then dehydrated with a graded acetone sequence (35, 50, 75, 95 % for 10 min, and 100 % for 15 min each). The samples were treated with a 1:1 acetone: resin mixture for 1 h, 1:3 mixture for 2 h, 100% resin overnight, and 100% resin for 2 h. After that, the samples were embedded in beam capsules with resin. The polymerization was then performed in an oven at 60 °C for 48 h. The specimen were divided into ultrathin sections with an ultramicrotome and a diamond knife. The sections were mounted on copper grids and treated for 15 min with 2% uranyl acetate and washed with double distilled water. The sections were then washed with double distilled water after being stained with lead citrate solutions for 10 min. Finally, a transmission electron microscope was used to examine the sections. 

### 4.8. Biofilm Breakdown Observations under Confocal Laser Scanning Microscope 

The membrane damage of the bacterial cells was determined and verified using the LIVE/DEAD BacLight (L7012) bacterial viability assays. CLSM was used in this study to investigate the effect of GEO on *Bg* biofilm formation. Each falcon tube contained 25 mL of MHB and was inoculated with 100 µL of the *Bg* suspension (1 × 10^6^ CFU/mL). The falcon tubes were then treated with the MIC concentration of GEO (12.5 µL/mL), while the control tube contained only MHB and *Bg* suspension. The control and treated falcon tubes were incubated for 24 h at 37 °C before being centrifuged at 10,000× *g* for 10 min. The pellet was then suspended in 2 mL of wash buffer (PBS solution). 1 mL of this suspension was added to 20 mL of PBS solution, and the tubes were incubated at room temperature for 1 h, with 15 min mixing intervals. After being centrifuged at 10,000× *g* for 10 min, the samples were resuspended in 20 mL of PBS solution. The pellets were then centrifuged again at 10,000× *g* for 10 min and resuspended in 10 mL of PBS solution.

Then, according to the manufacturer’s specifications, SYSTO 9 dye-3.34 mM and Propidium iodide-20 mM were mixed in a microcentrifuge tube in a 1:1 ratio. 3 µL of mixed stains were pipetted into milliliters of each sample and incubated for 15 min at room temperature. To ensure stain viability, stained samples were kept away from light throughout the staining process. The stained samples (10 µL) were pipetted onto glass slides and covered with coverslips. On the same day, CLSM was used to view the stained-glass slides.

### 4.9. Statistical Analysis

All experiments were performed in triplicate and data was expressed as mean value ± standard deviation. The data presented in this work were analyzed using a SAS 9.4 version of PROC ANOVA and mean significant differences have been detected at a probability level of 0.05 with the least significant difference (LSD). 

## 5. Conclusions

Natural antimicrobial agents are an important step toward reducing the negative effects of pesticides made from synthetic chemicals, such as slow biodegradation, toxic residues, resistance development in phytopathogens, and harmful effects on the environment, humans, and animals. Our findings demonstrated that ginger essential oil was primarily composed of a high percentage of monoterpenes and a low percentage of sesquiterpenes. The most abundant components in the hydrodistillation oil were Geranial, 1,8-Cineole, Neral, and Camphene. The in vitro analysis found that the ginger essential oil had significant antibacterial action against *B. glumae*.

Physicochemical characterization using vibrational spectroscopy analyses has revealed several functional groups and the chemical bonding for the AIs in GEO. It was also observed that the existence of benzene rings contributed by 1,8 Cineole and α-Zingiberene was determined by the Raman spectrum. Ginger essential oil antibacterial activity and inhibition of bacterial growth were dose-dependent. The treatment with ginger essential oil caused physical and morphological changes in the cell wall and membrane of *B. glumae*’s cells. Furthermore, the ginger essential oil was adsorbed, which resulted in cell wall damage or disintegration, and then penetrated the bacterial cells, causing intracellular leakage and cell death. The current findings have provided strong evidence that ginger essential oil is a strong antibacterial agent against *B. glumae*. However, further research should be performed to fully understand the control mechanisms to justify the practical applications of ginger essential oil in controlling bacterial diseases as a natural antibacterial agent.

## Figures and Tables

**Figure 1 plants-11-01466-f001:**
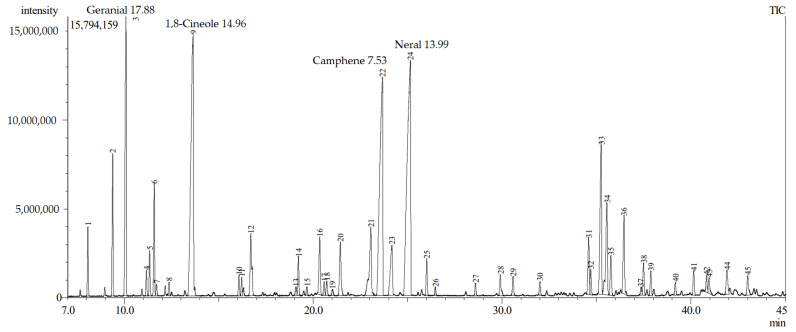
GC–MS chromatogram phytochemical analysis of ginger essential oil.

**Figure 2 plants-11-01466-f002:**
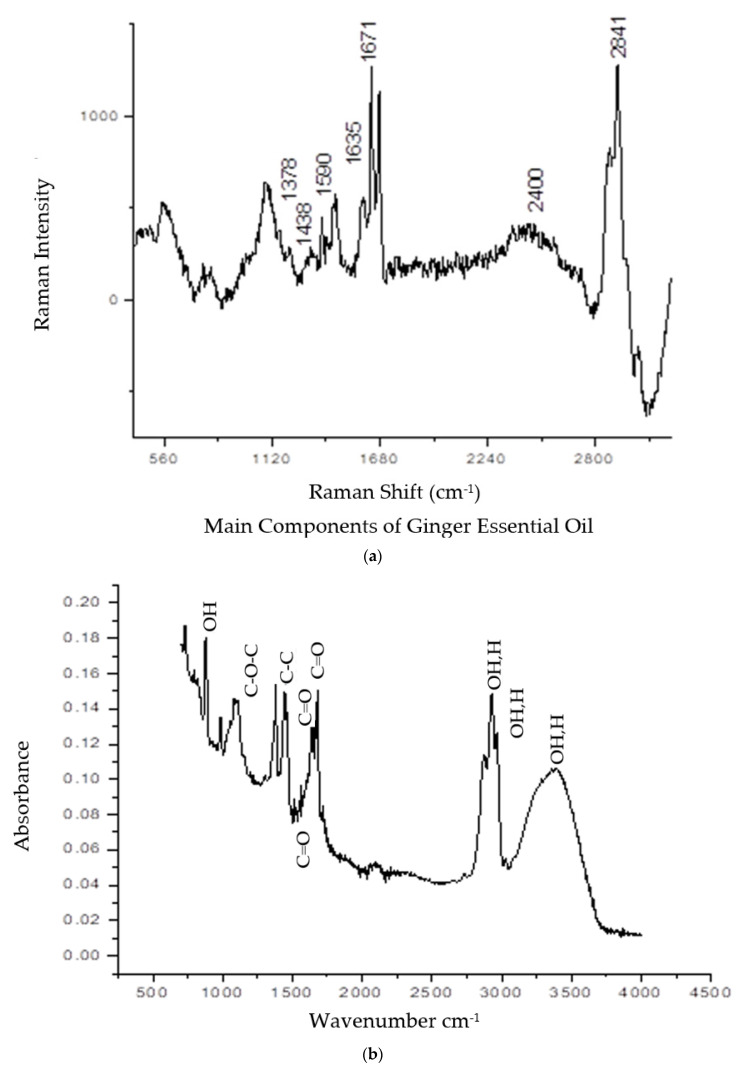
(**a**) The Raman spectrum (**b**) FTIR spectrum of ginger essential oil.

**Figure 3 plants-11-01466-f003:**
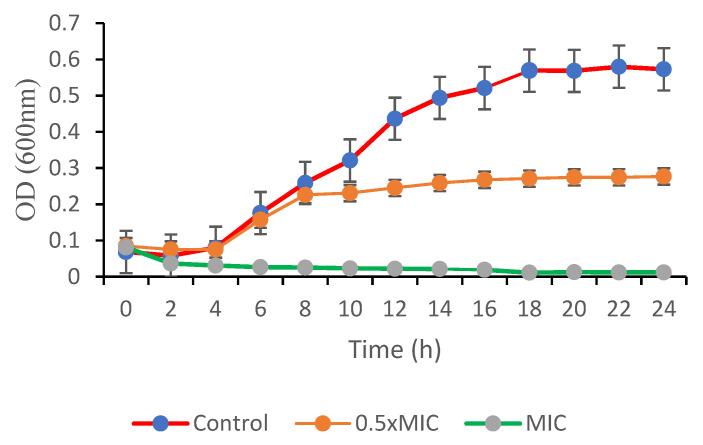
Kinetics of action of the GEO against *B. glumae.* The growth was recorded by quantifying the absorbance (O.D) at 600 nm hourly for 24 h. The error bars represent the standard deviations of three measurements.

**Figure 4 plants-11-01466-f004:**
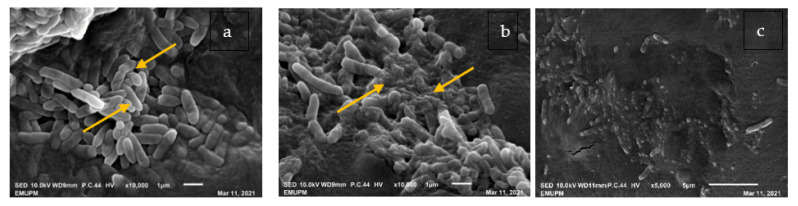
Scanning electron microscope images for the mechanism of action of GEO against *Bg.* (**a**) Regular rod shape for the DMSO-treated negative control of *Bg* culture after 24 h of incubation, Arrows show the smooth and regular cell shape (**b**) Irregular shape of *Bg* cells treated with GEO at MIC concentration, Arrows show the shriveled appearance the cell surface (**c**) Irregular growth of *Bg* cells treated with streptomycin (positive control) (10,000×).

**Figure 5 plants-11-01466-f005:**
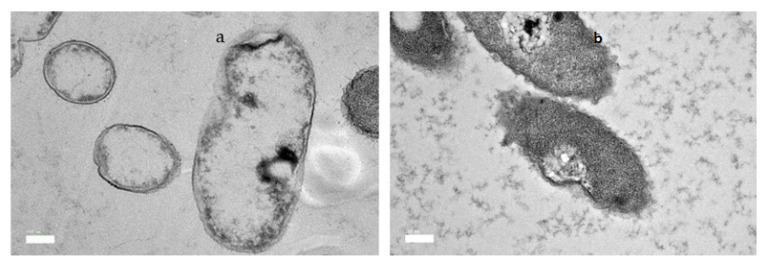
The mechanism of action of GEO against *Bg* was observed using a transmission electron microscope after being treated for 24 h. (**a**) *Bg* cells were treated with DMSO in normal rod shape (**b**) *Bg* cells were treated with GEO (MIC), causing the membrane cell to rupture.

**Figure 6 plants-11-01466-f006:**
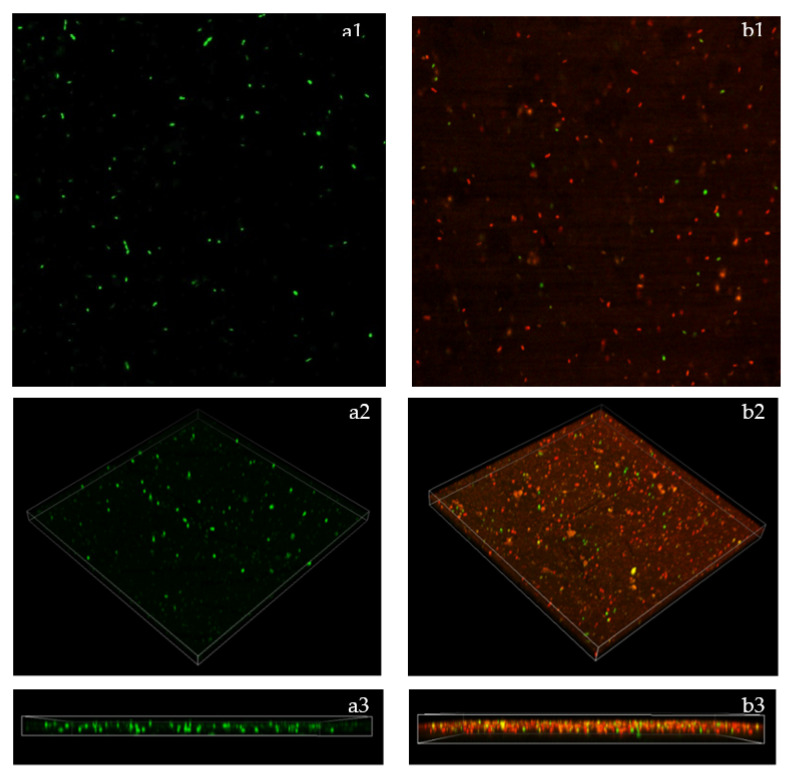
Confocal laser scanning microscope analyses were performed after 24 h treatment period. (**a1**) The green color of untreated *Bg* cells indicated that the cells were alive and undamaged. (**b1**) The image displays orthogonal 3D biofilm untreated *Bg* cells. (**a2**,**a3**) GEO treatment of *Bg* cells MIC resulted in red-stained cells, indicating that more dead cells occurred. (**b2**,**b3**) The image displays orthogonal 3D biofilm GEO treatment of *Bg* cells.

**Table 1 plants-11-01466-t001:** Chemical compounds in GEO using GC–MS by matching their retention time (Rt), retention index (RI), molecular formula, and mass spectral (MS). Data from the mass spectral databases FFNSC1.3.lib, NIST11.lib, and WILEY229. Lib.

	Chemical Compound	Rt (min)	RI (This Work)	RI [15]	RI [22]	Area (%)	Molecular Formula	MS
1	Heptan-2-ol	8.074	900	-	-	1.23	C_7_H_16_O	331
2	α-Pinene	9.386	933	937	922–955	2.76	C_10_H_16_	412
3	Camphene	10.087	951	943	933–962	7.53	C_10_H_16_	451
4	β-Pinene	11.173	978	-	959–986	0.52	C_10_H_16_	349
5	6-Methyl-5-hepten-2-one	11.344	983	-	-	0.95	C_8_ H_14_ O	347
6	Myrcene	11.588	989	975	973–993	2.31	C_10_H_16_	359
7	Sulcatol	11.706	992	-	-	0.25	C_8_H_16_O	332
8	α- Phellandrene	12.370	1007	987	989–1013	0.30	C_10_ H_16_	322
9	1,8-Cineole	13.646	1034	-	1005–1039	14.96	C_10_H_18_O	363
10	Terpinolene	16.074	1086	-	-	0.47	C_10_H_16_	408
11	Methyl lavender ketone	16.220	1089	-	-	0.44	C_10_H_20_O_2_	290
12	β-Linalool	16.705	1100	-	-	2.19	C_10_H_18_O	396
13	Camphor	19.081	1149	-	1045	0.26	C_10_ H_16_ O	299
14	Citronellal	19.224	1152	-	-	0.95	C_10_H_18_O	290
15	Isoneral	19.667	1161	-	-	0.22	C_10_H_16_O	407
16	Endo-Borneol	20.349	1175	-	-	1.70	C_10_H_18_O	355
17	Isogeranial	20.581	1179	-	-	0.34	C_10_H_18_O	404
18	4-Terpineol	20.719	1182	-	-	0.39	C_10_H_18_O	361
19	Cryptone	21.021	1188	-	-	0.19	C_9_H_14_O	361
20	α -Terpineol	21.442	1197	-	1159–1193	1.77	C_10_H_18_O	404
21	Citronellol	23.055	1231	-	1203–1229	3.27	C_10_H_20_O	358
22	Neral	23.666	1244	-	-	13.99	C_10_H_16_O	364
23	Geraniol	24.170	1254	-	1228–1258	2.35	C_10_H_18_O	364
24	Geranial	25.146	1274	-	1232–1267	17.88	C_10_H_16_O	342
25	2-Undecanone	26.014	1293	-	-	0.90	C_11_H_22_O	369
26	Methyl nonyl carbinol	26.466	1302	-	-	0.20	C_11_H_24_O	334
27	Citronellyl acetate	28.595	1348	-	-	0.31	C_12_H_22_O_2_	339
28	Geranyl acetate	29.911	1377	1383	1344–1385	0.63	C_12_H_20_O_2_	357
29	β-Elemene	30.585	1392	-	1370–1404	0.51	C_15_H_24_	385
30	Caryophyllene	32.007	1424	-	-	0.37	C_15_H_24_	353
31	Curcumene	34.588	1483	-	-	1.58	C_15_H_22_	423
32	Germacrene	34.698	1485	-	-	0.67	C_15_ H_24_	371
33	α-Zingiberene	35.247	1498	-	-	5.19	C_15_H_24_	312
34	α-Farnesene	35.551	1505	1433	1479–1518	3.51	C_15_H_24_	429
35	β-Bisabolene	35.766	1510	-	1485–1513	1.16	C_15_H_24_	396
36	β-Sesquiphellandrene	36.460	1527	1525	-	2.40	C_15_H_24_	418
37	Muurola-4,10(14)-dien-1.beta.-ol	37.348	1548	-	-	0.23	C_15_H_24_O	386
38	α-Elemol	37.494	1552	-	-	0.89	C_15_H_26_O	401
39	Nerolidiol	37.882	1561	1558	1535–1565	0.65	C_15_H_26_O	309
40	Sesquisabinene hydrate	39.174	1592	-	-	0.34	C_15_H_26_O	315
41	Zingiberenol	40.152	1617	1601	-	0.68	C_15_H_26_O	370
42	trans-Sesquisabinene hydrate	40.830	1634	-	-	0.49	C_15_H_26_O	394
43	Globulol	40.956	1637	-	-	0.51	C_15_H_26_O	402
44	Rosifoliol	41.911	1661	-	-	0.89	C_15_H_26_O	372
45	Shyobunol	43.007	1689	-	-	0.66	C_15_H_26_O	358

**Table 2 plants-11-01466-t002:** The detailed information on the peaks, wavelength, bonds, and functional groups for the FTIR spectrum of GEO.

Peaks Number	Wavelength cm^−1^	Chemical Bonds	Functional Compounds
1	3271	OH stretch;H-bonded	Carboxylic acid
2	2920	OH stretch;H-bonded	Carboxylic acid
3	2871	OH stretch;H-bonded	Carboxylic acid
4	1739	C=O stretch	Esters RCOOR
5	1678	C=O stretch	Carbonyl Compound
6	1640	C=O stretch	Carbonyl Compound
7	1590	C-C stretch	Aromatic stretch ring
8	1260	C-O-C	(R-O-R) Ether
9	870	OH bond	Phenol

**Table 3 plants-11-01466-t003:** Antibacterial activity of GEO at different concentrations and streptomycin against *B. glumae* using disc diffusion method.

	GEO Concentration (µL/mL)					DMSO	Antibiotic Streptomycin
	100	200	300	400	500		
Dimeter of zone of inhibition (mm)	7.1 ± 0.16 ^f^	8.0 ± 0.28 ^e^	10.6 ± 0.44 ^d^	14.3 ± 0.16 ^c^	15.1 ± 0.16 ^b^	0.0 ± 0 ^g^	24.1 ± 0.44 ^a^
Inhibition (%)	29.46%	33.19%	43.98%	59.33%	62.65%	0.0%	
Efficacy *	Not sensitive	Moderate sensitive	Moderate sensitive	Moderate Sensitive	Sensitive	Not sensitive	Extremely sensitive

^a–g^ Means with different letters differed significantly (*p* < 0.05). Values are mean ± standard deviation (n = 3). * Efficacy grouping = Not sensitive (total zone diameters of ≤8.0 mm), moderately sensitive (diameters between 8 and 14.0 mm), sensitive (zone diameters between 15 and 19 mm;), and extremely sensitive (zone diameters of ≥20 mm).

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
