# Peer review of "Phytochemicals Profiling, Antimicrobial Activity and Mechanism of Action of Essential Oil Extracted from Ginger (Zingiber officinale Roscoe cv. Bentong) against Burkholderia glumae Causative Agent of Bacterial Panicle Blight Disease of Rice"

_plants, 2022, doi:10.3390/plants11111466_

Round 1
Reviewer 1 Report
I suggest to the authors to improve the identification of compounds by calculating the retention index to be included in table 1. It would also be more correct for the percentage areas of the compounds to be defined using a GC / FID instead of the GC / MS.
Author Response
Reviewer 1:
I suggest to the authors to improve the identification of compounds by calculating the retention index to be included in table 1. It would also be more correct for the percentage areas of the compounds to be defined using a GC / FID instead of the GC / MS.
Response:
As recommended by the reviewer, the authors had added the retention index columns to table 1. “The retention index was calculated using the Kovats equation provided in 4.1 Material and Methods. The comparison of retention values of Sharma et al (2016) [16] and Babushok et al (2009) [24] was also provided in Table 1.” L 120 – 124
“According to Lucero et al (2009) [44] following equation was used to calculate retention indices of GEO.” L 398-409

Reviewer 2 Report
The paper submitted by Gunasena et al. is focused on the investigation of the chemical composition and antimicrobial activity of essential oil from ginger. The paper is in general well written. However some corrections and improvements are needed ti increase its scientific soundness and quality.
1) according to section 4.1 one essential oil was obtained and later investigated - in the title and other sections of the manuscript the authors use plural 'essential oils' - in my opinion this is not correct; usually one plant material contain one essential oils; this statement would be justified if authors investigated several oils from different plant materials or obtained from different batches of the same plant - please correct in the whole text
2) the documentation of the plant material used for the whole research is poor and needs improvement; the authentication method used for the material must be stated, together with person responsible for the identification, voucher specimens and source of the plant material - this is crucial is each study involving plants
3) fig 1 - please provide traditional structures of compounds not 3D models for the readers convenience
4) fig 2 - please label bands according to numbers from table 2
5) please improve the quality of fig 1 - chromatogram is poorly seen; labels are not clear for the reader
6) please provide MS data used for the confirmation of the compounds listed in table 1 - molecular formulas are not needed unless they are accompanied by HR-MS results
Author Response
Reviewer 2:
1.According to section 4.1 one essential oil was obtained and later investigated - in the title and other sections of the manuscript the authors use plural 'essential oils' - in my opinion this is not correct; usually one plant material contain one essential oils; this statement would be justified if authors investigated several oils from different plant materials or obtained from different batches of the same plant - please correct in the whole text
Response:
Authors have changed the word “essential oils” to “essential oil” in the manuscript as recommended by the reviewer. L 29, L 33, L 38, L 40, L81, L 85, L 89, L 100, L104, L 106, 108, L112, L 118, L 119, L 123, L 126, L 128, L 130, L 138, L 162, L 164, L167, L 170, L 171, L173, L184, L 188, L 189, L 191, L 192, L 195, L 196, L 199, L 200, L 201, L 203, L 206, L 208, L 213, L 226, L 229, L 235, L 236, L 239, L 241, L 245, L 246, L 247, L 254, L 255, L 265, L 267, L 268, L 270, L 272, L 283, L284, L 287, L 298, L 300, L 309, L 316, L 321, L 324, L 325, L 328, L 329, L 334, L 337, L 339, L 349, L 350, L 357, L 358, L 361, L 363, L365, L 372, L 375, L 376, L 394, L 409, L 410, L 413, L 423, L 425, L 432, L 437, L 438, L439, L 442, L 453, L 474, L 491, L 541, L 544, L 546, L 549, L 551, L 554
2) The documentation of the plant material used for the whole research is poor and needs improvement; the authentication method used for the material must be stated, together with person responsible for the identification, voucher specimens and source of the plant material - this is crucial is each study involving plants
Response:
The authors added the following sentence in the text for clarification on the status of ginger used in the work. “Fresh rhizome of ginger (Zingiber officinalae cv. Bentong) was obtained from Bentong, Pahang, Malaysia.” L 395- 396.
For your information in Malaysia ginger is a commercial crop. Two cultivars of ginger i.e. Zingiber officinalae cv. Bentong and Zingiber officinalae cv. Tambunan are the most popular and commercially cultivated in Malaysia. In my opinion, due to this ginger is a domestic and commercially planted so that, there is no need for the species identification by agriculture botanist.
3) fig 1 - please provide traditional structures of compounds not 3D models for the readers convenience
Response:
All 3D models were removed from the text as suggested by other reviewer.
4) fig 2 - please label bands according to numbers from table 2
Response:
The label bands were added in Figure 2(b) FTIR spectrum of ginger essential oil.
5) please improve the quality of fig 1 - chromatogram is poorly seen; labels are not clear for the reader
Response:
The authors have modified Figure 1 as recommended by the reviewer.
6) please provide MS data used for the confirmation of the compounds listed in table 1 - molecular formulas are not needed unless they are accompanied by HR-MS results
Response:
The authors believe that the molecular formulas are acceptable to be included in Table 1. The bioactive compounds have been confirmed by comparing with the published manuscripts such as Sharma et al (2016) [16], Babushok et al (2009) [23] and Adamu et al (2020) [28].

Reviewer 3 Report
The manuscript "Phytochemicals profiling, antimicrobial activity and mechanism of action of essential oils extracted from Ginger (Zingiber officinale R.) against Burkholderia glumae causative agent of bacterial panicle blight disease of rice" is interesting and has scientific merits. However, some points are required to be adjusted
Among the serious points:
1- The GC /MS identification is not accepted at all. Automatic matching the mass spectra with the reported data is not correct. Either Retention indices should be calculated and compared with what is published or Co-injection with authentic substances representing the major components
2- Quantification of the oil components should be better described
Minor points:
1- The use of FTIR has nothing to add since most of the components are known and this technique gives only an idea about the functional groups not the exact structures
2- The molecular structures of the identified compounds should be removed
3- The manuscript should be checked by an English native speaker to remove some typos and syntax
Author Response
Reviewer 3:
Among the serious points:
1- The GC /MS identification is not accepted at all. Automatic matching the mass spectra with the reported data is not correct. Either Retention indices should be calculated and compared with what is published or Co-injection with authentic substances representing the major components
Response:
The corrections were made in Table 1 as requested by the reviewer which includes the calculated retention indices along with some other comparisons. The bioactive compounds have been confirmed by comparing with the published manuscripts such as Sharma et al (2016) [16], Babushok et al (2009) [23] and Abdullahi et al (2020) [28].
2- Quantification of the oil components should be better described
Response:
The authors have accepted the recommendation and added some details to section 2.1 as recommended by the respected reviewer.
“The extraction yields were determined by dividing the amount of extracted oil by the amount of fresh material fed into the extractor. The extraction yields were estimated based on the results of several cycles. The results were presented as a percentage of extraction.”
L129-131
“Gas chromatography analysis had identified 45 peaks of active compounds available in GEO as shown in Fig. 1a. Besides, the retention indices of those active compounds can be calculated from the retention times using the Kovats equation. It also noticed that the major compounds were Geranial (17.88%), 1,8-Cineole (14.96%), Neral (13.99%), Cam-phene (7.53%), and α-Farnesene (3.51%) and their combination were more than 50%.” L 133 -138.
Minor points:
- The use of FTIR has nothing to add since most of the components are known and this technique gives only an idea about the functional groups not the exact structures
Response:
Authors accepted and highly appreciate the comment by the respected reviewer. FTIR is a valuable technique that can be used to identify and characterize unknown materials, such as functional groups, possible decomposition, and oxidation. It also can be used to detect possible contaminants and additives that are present in a material.
Indeed, the FTIR technique was used to evaluate the functional groups as in general information and those functional groups can not be assigned to a specific active compound in the ginger essential oil. However, it is understood that 5 major active compounds contributed about 57% content of the ginger oil as mentioned in the text. It is believed that the functional groups detected from the FTIR were mainly contributed by those 5 major compounds.
2- The molecular structures of the identified compounds should be removed
Response:
The authors have accepted the recommendation and had removed the molecular structures as suggested by the respected reviewer.
3- The manuscript should be checked by an English native speaker to remove some typos and syntax
Response:
The authors highly appreciate the comment from the respected reviewer. The English grammar has been corrected throughout manuscripts.

Round 2
Reviewer 2 Report
The manuscript was revised by the authors. However, in my opinion some corrections are still needed:
1) botanical authentication of the plant material is always obligatory; I understand that ginger in your region is a crop but proper documentation of the plant material used for the research is the only way to ensure repeatability of your experiments by other scientist in other part of the world.
2) I asked for MS data for all compounds included in table 1; if you provide molecular formula this should be supported with exact mass measured by MS
Author Response
RESPONSES TO REVIEWERS’ COMMENTS
Reviewer 2:
1. Botanical authentication of the plant material is always obligatory; I understand that ginger in your region is a crop but proper documentation of the plant material used for the research is the only way to ensure repeatability of your experiments by other scientist in other part of the world.
Response:
1. The plant specimen was identified by Dr. Khairil Mahmud of Institute of Bioscience, Universiti Putra Malaysia with the Voucher No: KM0006/22.
2. I asked for MS data for all compounds included in table 1; if you provide molecular formula this should be supported with the exact mass measured by MS.
Response:
2. As suggested by the reviewer, the details of the MS data for all bioactive compounds have added the Table 1.

Reviewer 3 Report
The manuscript has much improved and the authors amended all the required changes however the English need more work
Author Response
RESPONSES TO REVIEWERS’ COMMENTS
Reviewer 3:
1. The manuscript has much improved and the authors amended all the required changes however the English need more work.
Response:
1. Thank you very much for the comment. The manuscript was checked and corrected by English Editor as per highlighted in yellow colour.

Round 3
Reviewer 2 Report
No further comments.